# Low-carbon pathways for the booming express delivery sector in China

Peng Kang[1], Guanghan Song[1], Ming Xu [2,3✉], Travis R. Miller [4], Haikun Wang [5], Hui Zhang[6], Gang Liu [7], Ya Zhou [8], Junshu Ren[9], Ruoyu Zhong[10] & Huabo Duan [1✉]

Express delivery services are booming in both developed and emerging economies due to their low cost, convenience, and the fast growth in online shopping. The increasing environmental impacts of express delivery services and mitigation potentials, however, remain largely unexplored. Here we addressed such a gap for China, a country which is expanding online retail sales and express delivery rapidly. We found a total of 8.8 Mt of scrap packaging materials were generated by the express delivery sector in China in 2018. Its transportation-related GHG emissions surged from 0.3 Mt in 2007 to 13.7 Mt of $CO_2$-equivalent ($CO_2e$) in 2018, with an average of 0.27 $kgCO_2e$ per piece. Over 80% from online shopping deliveries. We predict these emissions will reach 75 $MtCO_2e$ by 2035. Nevertheless, it is possible to mitigate such GHG emissions by 102~134 $MtCO_2e$ between 2020 and 2035 if a suite of policies is adopted, including a slowdown of delivery speed, fuel system upgrades, packaging materials reduction, logistics optimization, and carbon pricing.

[1] College of Civil & Transportation Engineering, Underground Polis Academy, Shenzhen University, Shenzhen 518060, China. [2] School for Environment and Sustainability, University of Michigan, Ann Arbor, MI 48109-1041, USA. [3] Department of Civil and Environmental Engineering, University of Michigan, Ann Arbor, MI 48109-2125, USA. [4] Department of Chemical and Environmental Engineering, Yale University, New Haven, CT 06511, USA. [5] Joint International Research Laboratory of Atmospheric and Earth System Sciences, School of Atmospheric Sciences, Nanjing University, Nanjing 210023, China. [6] School of Chemistry & Environmental, Wuhan Institute of Technology, Wuhan 430205, China. [7] SDU Life Cycle Engineering, Department of Chemical Engineering, Biotechnology, and Environmental Technology, University of Southern Denmark, Odense 5230, Denmark. [8] Institute of Environmental and Ecological Engineering, Guangdong University of Technology, Guangzhou 510006, China. [9] SF Group, Beijing 100010, China. [10] China Center for Special Economic Zone Research, Shenzhen University, Shenzhen 518060, China. ✉email: mingxu@umich.edu; huabo@szu.edu.cn

Express delivery is one of the fastest-growing industries worldwide, especially in emerging economies. The majority of express delivery services are associated with online shopping[1,2], which is also growing rapidly throughout the world (33% annually by number of transactions and 34% by value) (see Supplementary Figs. 1 and 2)[3,4]. The growth of online customer-to-customer (C2C) trading platforms where individuals sell items to each other, such Taobao in China and eBay in the US, has led to the rapid growth to a rapid growth in C2C express delivery services.

In China, online retail sales grew 25% annually over the past decade[4], reaching $1.1 trillion USD in 2018. This represented 18% of all retail sales in China, and 15% of the global online retail sales—more than the US and the UK combined[5,6]. The boom in China's online retail business has been enabled by the rapid development of express delivery (see Supplementary Figs. 3 and 4)[7,8]. China's express delivery market recorded over 50.7 billion orders in 2018 and 63.5 billion in 2019, with around 80% of the deliveries due to online shopping[9]. The volume of China's express deliveries is approximately 60% of the global total and triple that of the US[10,11]. Most express delivery in China occurs in highly populated and developed regions, such as the Pearl River Delta in South China, the Yangtze River Delta in East China, and the Beijing–Tianjin–Hebei region in the North China Plain (see Supplementary Fig. 5)[4]. Our business-as-usual (BAU) projection estimates that the annual express deliveries will reach over 200 billion in China by 2035 `(see Supplementary Fig. 6).

Despite the convenience express delivery offers, authorities and consumers increasingly expect the express delivery sector to account for and mitigate its environmental burdens, such as packaging material waste generation and transportation-related greenhouse gas (GHG) emissions[12,13]. While studies on packaging material waste have increased in the literature[14,15], there has been little information on the climate impacts of logistics and transportation. Although online shopping can in theory reduce environmental impacts by avoiding trips to brick-and-mortar stores[16,17], the reality is that consumers are hardly offsetting the trips through online shopping[18,19]. On the contrary, online shopping is, to a large extent, a net addition to conventional shopping[20]. To date, the scale and trend of environmental impacts from the express delivery industry in China remain largely unknown, and the same concern exists in many other countries as well.

Accounting for GHG emissions is a prerequisite for greening the express delivery service. Based on field surveys and robust modeling for years 2007 to 2018, in this study to the best of our knowledge for the first time to the best of our knowledge, estimated the generation of scrap packaging materials and quantify the GHG emissions from logistics and transportation of express deliveries in China from 2007 to 2018. We also projected future GHG emissions up till 2035 under various scenarios and identified effective policy options to mitigate these emissions (see Supplementary Fig. 7).

## Results and discussion

**Parcel shipment and weight**. In 2018, approximately 50.7 billion parcels were delivered in China, over 80% of them from online shopping. By sector, 85%, 10%, and 5% of these parcels were delivered via road, air, and railway transportation, respectively; and 75%, 23%, and 2% were intercity, intracity, and international deliveries, respectively[3]. Our surveys show that most of these deliveries used corrugated boxes (53.5%), followed by plastic bags (33.5%), envelopes (5.4%), and other types of packaging (7.6%) (see Fig. 1). The volumes of the top three most frequently used containers were as follows: small corrugated boxes (10.6 billion), large plastic bags (8.0 billion), and medium-sized corrugated boxes (7.8 billion) (see Supplementary Fig. 8). The average weight of a parcel, including the weight of the goods, was approximately 1 kg, ranging from 0.1 kg for envelopes to 4.0 kg for oversized

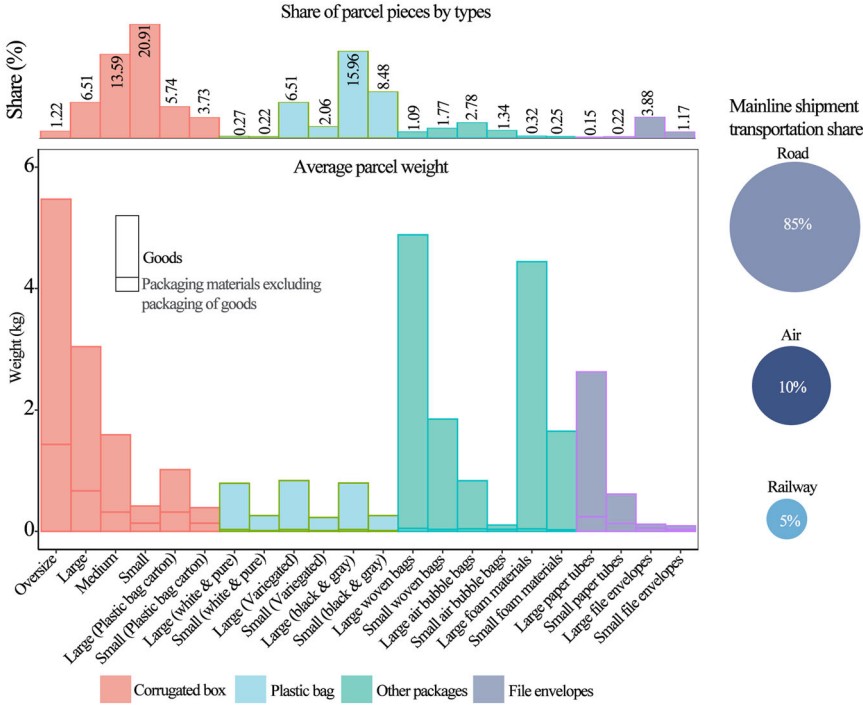

**Fig. 1 Express deliveries in China in 2018 by parcel type and transportation mode.** Four types of packaging materials of parcels were classified: corrugated box, plastic bag, file envelopes, and other packaging. The ratios of the categories of packaging type were calculated. Sample batches of each packaging material category, of different sizes and materials, were weighed, and the mean values calculated. If divided by sectors, 85%, 10%, and 5% of these parcels were delivered via road, air, and railway transportation, respectively.

corrugated boxes. Notably, packaging materials for parcels, excluding the packed goods inside, exceeded 20% (on average) of the total parcel weight.

Since a majority of packaging materials for parcels are single-use packaging, we estimate that a total of 8.8 ($\sigma = 0.8$) million metric tons (Mt) of scrap packaging materials from the express delivery sector were generated in 2018. This accounts for approximately 3.9% of China's total municipal solid waste (MSW) volume. Specifically, corrugated boxes, plastic bags, envelopes and other packaging containers accounted for 90.8%, 5.9%, 1.7%, and 1.6% by weight, respectively. While the mass fraction for recycled corrugated boxes was around 80%, the recycling rate for plastic bags was only about 1.5% (by weight). Unrecycled packaging materials were collected with MSW ending up in landfill sites (57%), incinerators (41%), and open dumps (2%)[14].

**GHG emissions from parcel deliveries**. We estimated that 13.7 ($\sigma = 2.2$) Mt of $CO_2$-equivalent ($CO_2$e) of GHG emissions were caused by the logistics and transportation of express deliveries in China in 2018, over 80% from online shopping parcels and nearly 99% from intercity deliveries. These GHG emissions accounted for approximately 7% of the total GHG emissions from the entire logistics industry in China[21]. Total GHG emissions varied with

parcel size and packaging materials, ranging from 10 to 2800 kilo-tonnes (kt) of $CO_2$e (see Supplementary Fig. 9). The average GHG emissions per delivery piece are 0.27 kg$CO_2$e.

Total GHG emissions from express deliveries by region of origin ranged from 50 to 1800 kt$CO_2$e in 2018 (see Supplementary Fig. 10), with the majority originating from affluent, densely populated coastal regions. Specifically, the GHG emissions of express deliveries originating from Guangzhou and Shenzhen in South China and Jinhua in East China exceeded 1000 kt$CO_2$e (see Supplementary Figs. 11 and 12). In contrast, GHG emissions from many Central and West China regions were less than 100 kt (see Supplementary Figs. 13 and 14). Overall, emissions associated with the three key economic high-population centers (Yangtze River Delta, Pearl River Delta, and Beijing–Tianjin–Hebei, and these three regions account for 24% of the national population) respectively accounted for 35%, 26%, and 8% (70% in total) of the national total.

GHG emissions from developed regions in the coastal areas were much higher than those from less developed areas (see Fig. 2). Among all 42 regional centers, two megacities—Guangzhou and Shenzhen in South China—contributed the largest amounts (20%) of GHG emissions, as origins of express deliveries: 1683 kt and 968 kt$CO_2$e, respectively.

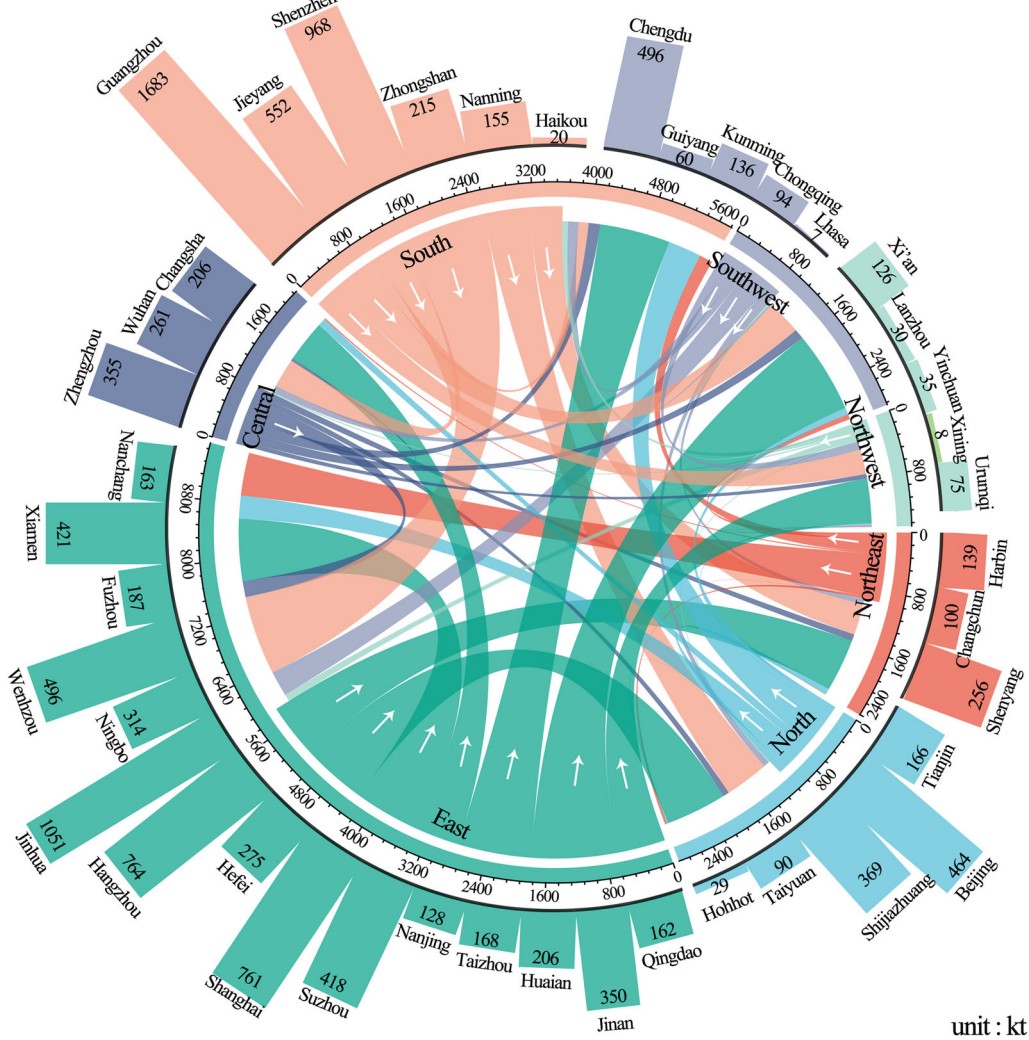

**Fig. 2 Flows of GHG emissions from the logistics and transportation of intercity express deliveries among the 42 centers. (Aggregated emissions data of Phases I–V).** Arrows show the direction of deliveries from origin (sender) region to destination (receiver) region. The emission amounts are attributed to the origin regions.

While intercity deliveries accounted for 75% of the total number of parcels delivered in 2018, they contributed over 99% of the GHG emissions (see Supplementary Fig. 15). Specifically, there are five phases in the intercity delivery process (see Supplementary Fig. 16, as a sample case): (1) Phase I, pick-up within the origin city and transportation to the distribution center; (2) Phase II, transportation from the distribution center to the regional center; (3) Phase III, mainline transportation from the regional center of the origin to the regional center of the destination; (4) Phase IV, transportation from the regional center to the destination city; and (5) Phase V, delivery within the destination city (from distribution center to customer). Among these five phases, Phase III accounted for the longest distances and the vast majority (95%) of the total GHG emissions from express deliveries (13 MktCO$_2$e in 2018, see Supplementary Fig. 15). The GHG emissions of Phase IV (regional center to destination city) are higher than those of Phase II (origin city to regional center). That is due to the majority of parcels originating from coastal areas where the distance from origin city to regional center is generally shorter than that from regional center to destination city; destination cities are often scattered across China rather than being clustered around large urban areas (see Supplementary Figs. 17 and 18). In contrast, the GHG emissions of Phase I were greater than those in Phase V, because the average intracity distance in the origin cities in coastal regional centers is longer than those of the smaller destination cities in less urbanized regions (see Supplementary Fig. 19).

Taking intercity express deliveries for example, we examine the correlations between the GHG emissions of express deliveries and a variety of socioeconomic factors. There is a significant positive correlation between the GHG emissions and the rate of urbanization (share of urban population) or GDP (see Supplementary Figs. 20 and 21). However, the average GHG emissions per parcel for each city and the urbanization rate or GDP are negatively correlated, likely due to more efficient logistics and transportation systems in more affluent and urbanized areas (see Supplementary Figs. 22 and 23).

We further conducted a stepwise regression by removing six less important variables, as indicated in Supplementary Fig. 24, including GDP per person, average delivery time, and expenses per customer (see Supplementary Table 1). The results identify several important factors for GHG emissions per parcel delivered (see Table 1 and Supplementary Table 2), including road travel distance per parcel, urbanization rate, and delivery volume per customer. These three factors are able to explain 94% of the variance of GHG emissions of express deliveries, which is due to that the key parameter for this method is the transportation distance, especially the road transportation distance per piece.

**Scenario-based projections and GHG mitigation strategies.** From 2007 to 2018, GHG emissions from logistics and transportation of express deliveries in China increased from 0.3 Mt to 13.7 ($\sigma = 2.2$) MtCO$_2$e (see Supplementary Fig. 25). Logistic

growth model can be considered a reasonable approach to describe the growth in the volume within growth threshold, such as urbanization, logistics, and so on[22,23]. Based on a logistic growth model, the number of express deliveries per person per year will increase from 45 pieces in 2019 to approximately 160 in 2035 (see Supplementary Fig. 6). Combined with population projections for China, the total volume of express deliveries will increase to approximately 220 billion in 2035 under a BAU scenario. As a result, GHG emissions of express deliveries are expected to grow at a constant rate from 2019 to 2027 and then more slowly until 2035. In the BAU scenario, GHG emissions of express deliveries are expected to reach approximately 75 MtCO$_2$e yr$^{-1}$ by 2035, an amount equal to the GHG emissions from fuel combustion in many countries such as Israel and Austria in 2016[24].

In addition to the BAU scenario, we examine the potential of a suite of strategies on reducing GHG emissions of express deliveries through other scenarios, which align well with the current status and future directions of greening the express delivery sector (see Table 2 in Methods and Supplementary Table 3). These strategies include: faster delivery speed by using more air freight (S1) or slower using more rail freight (S2); reducing the GHG emission intensity of delivery through the use of low-carbon fuels or improvement of fuel economy (S3); lightening parcel weight by using less packaging material (S4); optimizing the logistics system to reduce delivery distance (S5); and combining measures S2–S4 for the final scenario (S6). Our results show that both S3 and S4 can achieve a slight emissions reduction compared to the BAU, while S2 and S5 can lead to a much larger reduction, of approximately 100 MtCO$_2$e, between 2020 and 2035. Among all the effective strategies, delivery companies are most motivated to optimize their logistics system, to reduce costs (S5)[25]. The other three strategies (S2–S4), collectively represented by S6 (with a combined accumulated reduction of 134 MtCO$_2$e), rely more on policy intervention and market mechanism driven by consumer demand.

The scenario analysis has several important policy implications. First, fuel-based strategies (S3) and weight reduction through reducing the amount and weight of packaging materials (S4) have limited potential to reduce GHG emissions from express deliveries, especially for intercity deliveries, although they could be effective for intracity deliveries. Similarly, the reduction of packaging materials (S4) can hardly deter the increasing trend of GHG emissions, while it can greatly reduce packaging waste generation[14,26].

Second, slower delivery (S2) for intercity delivery has a larger potential to reduce emissions, and can be achieved by avoiding air freight and replacing road transportation by railway transport (or inland waterways) for Phase III. In contrast, continuous promotion of faster delivery (S1) can significantly increase emissions. As shown in Fig. 3, increasing delivery speed via a 1-day reduction in transportation time, by using more air freight to replace road transportation for long-distance intercity deliveries, will increase the unit cost of delivery by a small

**Table 1 Stepwise regression for GHG emissions from express deliveries with various factors.**

| Index | Estimated coefficient | Std. Error | t value | Significance level P |
|---|---|---|---|---|
| Intercept | 2.25E + 00 | 4.19E − 01 | 5.379 | 4.04E − 06[**] |
| Road transportation distance per piece ($x_1$) | 2.15E − 03 | 1.00E − 04 | 19.908 | <2E − 16[**] |
| Urbanization ratio ($x_2$) | −2.35E + 00 | 5.89E − 01 | −3.983 | 2.97E − 04[**] |
| Delivery volume per person ($x_3$) | 1.7E − 03 | 8.43E − 04 | 2.117 | 4.08E − 02[*] |

$R^2 = 0.9439$, Adjust. $R^2 = 0.9394$, $F(4) = 213$, significance level $P < 2.2E − 16$.
[*]$p < 0.05$.
[**]$p < 0.001$.

**Table 2 Scenarios for future projection of express delivery in China.**

| Scenario | Description | Note |
|---|---|---|
| S1 (faster delivery) | The proportion of air mode was increased by 10% (to substitute for road freight). | Considered to speed up the delivery service based on examining future directions of express delivery sector[38]. |
| S2 (slower delivery) | The proportion of railway mode was increased by 15% (to substitute for road freight). | Considered a slowdown of delivery speed (a hypothetical scenario which is probably contrary to actual demand). |
| S3 (fuel standard upgrade) | More stringent emission standards for heavy-duty vehicles came into effect (from current China IV and V standards to China VI)[a]. | More stringent standards were drawn up leading to the implementation of China VI standards in 2020[39]. |
| S4 (less packaging material) | The volume of packaging materials was reduced by 15%. | An enforceable requirement for the reduction of packaging materials in 2019 according to the "Express Delivery Interim Regulations" promulgated by the State Council of the People's Republic of China[b][40]. |
| S5 (logistics optimization) | The transportation distance (Phase III) for each parcel was reduced by 15% on average. | According to the practical case of the top three domestic private express delivery companies (major express companies in China) in 2018[25]. |
| S6 (multiple measures combined) | An all-encompassing scenario to combine multiple measures (S2–S4) | |

[a]The Chinese Emission Standards (China I–VI) were published together by the MEP and SAC, and they comply with the European Emission Standards (Euro I–VI).
[b]Using electronic waybill, reusable boxes and bags, recycling and green packaging, and so on.

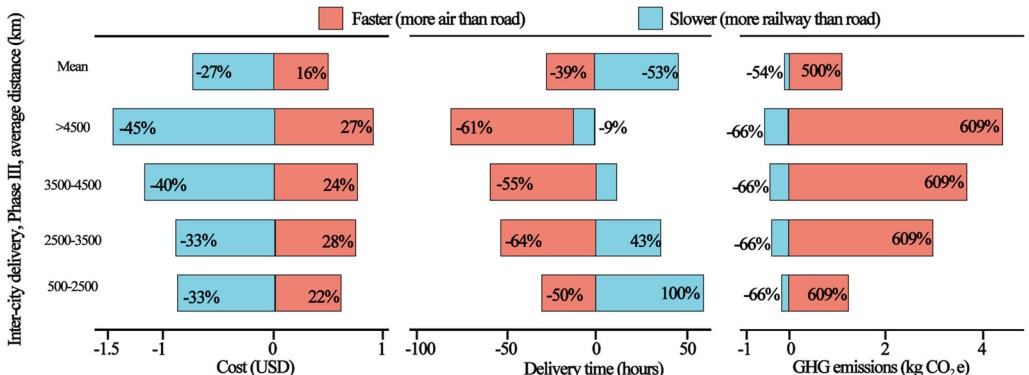

**Fig. 3 Tradeoffs in average cost, delivery speed, and GHG emissions for delivering one piece.** Data on average delivery time and fees for different transportation modes are from several major express delivery companies in China.

amount—approximately $0.48 USD per parcel, but significantly increase GHG emissions, by 1.0 kgCO₂e (4 times higher) per delivery on average (see Table 2 in Methods and Supplementary Table 3). On the other hand, decreasing delivery speed by adding two days of transportation time by using more railway shipping, while it will reduce the delivery cost by another relatively small amount—approximately $0.72 USD per parcel, will decrease GHG emissions by 0.12 kgCO₂e (one-half) per delivery. These tradeoffs are also evaluated in Fig. 3 for five delivery distance intervals, all of which have similar transportation times and costs: <500 km, 500–2500 km, 2500–3500 km, 3500–4500 km, and >4500 km (see Supplementary Table 4). By extrapolating these estimates to the projected deliveries up to 2035, using more railway shipping (Scenario S2) will reduce GHG emissions by 80 ktCO₂e in 2018 and 290 ktCO₂e in 2035, with the "expense" of longer delivery times (by 2 days on average) (see Supplementary Fig. 26). This can be achieved by offering multiple delivery choices with price incentives to encourage consumers to choose slower delivery for noncritical parcels. Carbon pricing strategies such as a carbon tax can also be considered, to make slower delivery more competitive with the marginally increased cost for faster delivery (see Supplementary Fig. 27 and Table 5).

Based on our field surveys and modeling analysis, we estimate that a total of 8.8 Mt of scrap packaging materials were generated in 2018, which accounted for approximately 4% of China's total

MSW. The transportation-related GHG emissions generated by the express delivery sector in China surged from 0.3 in 2007 to 13.7 MtCO₂e in 2018. There is a significant positive correlation between the GHG emissions and the rate of urbanization. GHG emissions associated with the three key economic high-population centers (Yangtze River Delta, Pearl River Delta, and Beijing–Tianjin–Hebei) accounted for approximately 70% of the national total.

We also project future GHG emissions up till 2035 with a peak at 75 MtCO₂e, subject to a logistic growth model and scenario analysis. The scenario analysis has several important policy implications: First, moderate fuel-based strategies (S3) and weight reduction through lessening packaging materials (S4) have limited potential to reduce GHG emissions from express deliveries. Second, slower delivery (S2) for intercity packages has a larger potential to reduce emissions. Considering that the upcoming interval (from 2020 to 2035) is the key phase of the development of the express delivery sector, the suite of policies for emission reduction, including a slowdown of delivery speed, fuel system upgrades, packaging materials reduction, logistics optimization, and carbon pricing, are important to be considered by relevant industry and governmental agencies.

Our study is provides a first estimate of GHG emissions of the express delivery industry in China to the best of our knowledge. Limitations exist and represent important future research needs.

First, it is necessary to conduct more extensive and carefully designed surveys to better understand key characteristics of the rapidly developing express delivery industry. Second, the logistic growth model for future parcel demand can be further improved by considering more factors (e.g., regional and temporal differences within China). Third, our study covers most key phases of the express delivery supply chain; but other aspects such as warehousing, international delivery service, packaging materials, and alternatives for last-mile delivery service (e.g., by drones)[16] can also affect GHG emissions and need to be included in future study. Fourth, other environmental impacts aside from GHG need to be considered. Local air pollution such as particulate matter from transportation also causes public health issues and needs to be addressed. Lastly, the environmental impacts of an industry or a company probably strongly correlate with firm-level corporate social responsibility (CSR) efforts[27,28]. An in-depth analysis of how various CSR initiatives or measures can affect the GHG emissions of the express delivery industry will be useful to guide companies to develop appropriate CSR strategies.

## Methods

**Scope of this study.** In accordance with the definition of the Postal Law of the People's Republic of China[29] and the World Trade Organization[30], express delivery refers to a delivery activity (only for parcels and documents of a given size and weight) which is finished promptly within a specified time frame, including online shopping delivery service (e-commerce to customer or business to customer), customer to customer, and business to business deliveries. This implies that logistics and transportation for both large freight (business to customer and business to business) and industrial or business freight (business to business) are not considered express delivery, and are therefore not included in this study. Specifically, express delivery is relatively independent from the national postal service, and can have dedicated sorting, warehousing, and delivery processes, and independent equipment, facilities, delivery channels, and personnel. We define express delivery as the delivery service of both intracity and intercity parcels. In addition, we only focus on GHG emissions, measured by $CO_2$-equivalents ($CO_2e$) from the transportation phase of domestic express deliveries, since transportation is the dominating stage of energy consumption and GHG emissions for delivery services[31]. The system boundary of this study is depicted in Supplementary Fig. 28.

**Estimate of the scrap packaging materials and quantification of the transportation impact.** Since a majority of outward protective packaging categories for shipping are single-use packaging, we can straightforwardly estimate the generation of scrap packaging materials in terms of the consumption of different types of materials. The reuse rate only accounted for approximately 5% by weight in 2018.

We utilize the standard EN 16258:2012 (Methodology for Calculation and Declaration of Energy Consumption GHG Protocol Product Lifecycle Accounting and Reporting Standard), the most accepted standard globally[32], to quantify GHG emissions. There are two types of GHG emissions involved: direct emissions from transportation-related combustion of fuels (mobile sources) (Scope 1), and indirect GHG emissions attributable to the consumption of electricity (Scope 2). More details on the determination of method and data sources can be found in the Supporting Information.

Specifically, the following steps were taken to calculate GHG emissions:

STEP 1: Determine the express delivery volume (parcel pieces) in China in a given year (y) (Eq. 1):

$$D_y = D_{c,y} + D_{u,y} \tag{1}$$

where $D_y$ is the total express delivery volume in a given year; $c$ indicates the intercity express delivery service, and $u$ stands for intracity express delivery service.

STEP 2: Determine the types of parcels according to their packaging materials and size, and then determine the average weight of each type of delivered parcel (Eq. 2) and its packaging materials (Eq. 3) from field surveys.

$$W_p = \sum_{i=1}^{n} Wp_i * pt_i \tag{2}$$

$$W_{pm} = \sum_{i=1}^{n} Wp_i * pt_i \tag{3}$$

where $W_p$ and $W_{pm}$ represent the weight of the parcel and its packaging materials, respectively; parcel types 1:$n$ are indexed by $i$, and $pt_i$ stands for the share of type $i$ parcels.

STEP 3: Estimate the generation of scrap packaging materials for express delivery service in China in a given year (y) (Eq. 4).

$$S_{pm,y} = \sum_{i=1}^{n} Wpm_i * pt_i * (1 - rt_i) \tag{4}$$

where $S_{pm,y}$ represents the weight of scrap packaging materials generated, and $rt_i$ stands for the reuse rate of type $i$ parcels.

STEP 4: Determine the transportation mode and distance of intracity express delivery. Road freight by truck is the main transportation mode for intra-city deliveries. Considering the fragmentation and hierarchy of the urban ecosystem in China, it can be assumed that the urban form was configured as encompassing one or more centers and a few satellite patches. The distance for intracity express delivery includes three parts (see Supplementary Fig. 29): (1) the distance across each patch (the buildup area of human activities in the urban environment); (2) the distance between the center (the core area of human activities in the urban environment) and the other patches; and (3) the distances between the other patches. The first part is computed using the radius and the compactness ratio. The assumption is that a single patch can be viewed as a circle, then adjusted with the applicable landscape form index using Eqs. 5–7. Conversely, the other two parts are based on the mean Euclidean distance between any two other patches. As shown in Supplementary Table 6 (validation analysis), this method can be used to accurately estimate intracity delivery distance.

$$Rrd_k = \sqrt{\frac{A_k}{\pi}} \tag{5}$$

$$CoI_k = 2\frac{\sqrt{\pi A_k}}{P_k} \tag{6}$$

$$Td_{fir} = \sum_{i=1}^{n} \frac{pop_i * A_i}{\sum_{1}^{n} pop_i * A_i} * Rrd_k * \left(\frac{2}{1 + 2*CoI_k}\right) \tag{7}$$

where $Td_{fir}$ represents the delivery distance for a single patch; $CoI_k$ is the compactness ratio of patch $k$, ranging from 0 to 1 with a higher value indicating a more compacted shape and a value closer to 1 indicating a shape closer to a circle; $pop_k$ is the population density for patch $k$; $P_k$ represents the perimeter of patch $k$'s contour; $A_k$ represents the area of patch $k$; and $\sum_{1}^{n} pop_i * A_i$ is the total population of all the patches.

STEP 5: Determine the transportation mode and distance of intercity express delivery. We use a hub-and-spoke model to represent the intercity delivery system which is made up of a small number of hubs, with many spokes within each hub (regional center) covering a number of spokes (the cities it covers) (see Supplementary Fig. 16 and Fig. 30). Specifically, we follow three steps to estimate the distance:

First, we identify 42 regional centers for intercity express delivery (see Supplementary Figs. 31 and 32) according to the hub-and-spoke model[33,34] and for our analysis of 21,000 express delivery waybills. Secondly, we examine 500 randomly selected waybills originating from one particular regional center, to estimate the share of express deliveries by each destination regional center. We develop a 42*42 matrix of these shares with rows and columns respectively representing regional origin and destination centers (see Supplementary Fig. 33 and Table 7). Thirdly, we assume that 85%, 10%, and 5% of intercity parcels are transported via road, air, and railways, respectively, based on actual aggregated statistical data[4]. We also categorize inter-city deliveries into four groups based on the delivery distance, and qualitatively estimate the share of air freight for deliveries in each group: ≤500 km (none by air), between 500 and 1000 km (a few by air), between 1000 and 2000 km (moderate number by air), and ≥2000 km (majority by air)[34]. We then assume that the shares of deliveries by air and by railways substitute road 2 and 1% per distance interval (see Supplementary Fig. 34).

STEP 6: Calculate transportation-related GHG emissions of intracity express delivery. Road freight by truck is the main transportation mode for intra-city deliveries. Electric bicycles are used for last-mile pick-up and distribution service. The GHG emissions of intracity express delivery can then be calculated in Eq. 8:

$$Ce_{ci,y} = W_{p,i} * [FT_c * (Td_{ci} - DL) + (FE_c * Dl)] * D_{u,y} \tag{8}$$

where $Ce_{ci,y}$ is the GHG emissions of intracity express delivery; $Wp$ is the weight of parcels; $Td_{ci}$ is the transportation distance of the intracity express deliveries; $Dl$ is the last-mile delivery distance; $FT_c$ is the GHG emissions factor (scope 1) of truck delivery; and $FE_c$ is the GHG emissions factor of the electric three- or two-wheelers (scope 2).

STEP 7: Calculate transportation-related GHG emissions of inter-city express delivery service. Regarding intercity deliveries, there are three different types of transportation modes (see Supplementary Fig. 16 and Fig. 31): (1) receiving and distribution service within a city, including Phases I and V, which are similar to those for intracity deliveries; (2) branch line transportation between the city and the regional center, including Phases II and IV; (3) mainline transportation (Phase III) between regional centers, which is the most complex process.

(1) Phases II and IV: Road transportation by truck is the major mode for Phases II and IV, whose distances can be determined by the publicly available but professional map software (i.e., Baidu Map)[35]. The GHG emissions of these phases

can be calculated by Eq. 9.

$$Ce_{bl,y} = Wp * Td_{bl} * FT_c * D_{c,y} \qquad (9)$$

where $Ce_{bl,y}$ is the GHG emissions of Phase II or IV; $Wp$ is the weight of the parcel; $Td_{bl}$ is the transportation distance for Phase II or IV; and $FT_c$ is the GHG emissions equivalent factor of the truck.

(2) Phase III: Multiple transportation modes are used in Phase III, whose distances were also determined by the publicly available but professional map software (i.e., Baidu Map). The GHGs emissions of this phase can be calculated by Eq. 10.

$$Ce_{ml,y} = \sum_{j=1}^{n} \sum_{i=1}^{m} Tr_j * Wp * DL_m * FT_{cm} * OD_{pm} * D_{c,y} \qquad (10)$$

where $Ce_{ml,y}$ is the GHG emissions of Phase III; $Wp$ is the weight of the express parcel; $Pm$ is the proportion of the transportation type (road, air, and railway); $Dt_m$ is the distance of the corresponding transportation type; $FT_{cm}$ is the GHG emissions equivalent factor of the corresponding transportation type; and $OD_{pm}$ is the transferring proportion from origin to destination.

(3) Total transportation-related GHG emissions for inter-city express delivery service. The aggregated GHG emissions of intercity express delivery service can be calculated by Eq. 11.

$$Ce_{c,y} = Ce_{ci_{I,y}} + Ce_{bl_{II,y}} + Ce_{ml,y} + Ce_{bl_{Iv,y}} + Ce_{ci_{v,y}} \qquad (11)$$

where $Ce_{c,y}$ is the GHG emissions of intercity express delivery service; and $Ce_{ci\_I}$, $Ce_{bl\_II}$, $Ce_{ml}$, $Ce_{bl\_IV}$, and $Ce_{ci\_V}$ are the emissions of Phases I–V, respectively.

STEP 8: Calculate the total transportation-related GHG emissions of both intercity and intracity express delivery service. Equation 12 expresses the sum of the emissions of intracity and intercity express delivery services.

$$Ce_{t,y} = Ce_{ci,y} + Ce_{c,y} \qquad (12)$$

where $Ce_{t,y}$ is the total transportation-related GHG emissions of the express delivery sector.

**Data sources.** Multiple data sources were employed to understand the express delivery volume, the characteristics of logistics and transportation (e.g., mode and distance), and GHG emission factors.

The volume of express delivery parcels is sourced from various official statistics (see Supplementary Tables 7 and 8). Data include the volume, in both pieces and monetary values, of intracity and intercity deliveries, domestic and international deliveries, and interprovince deliveries.

We consider four types of parcels according to the Express Service National Standard (GB/T 27917.1-2011), including corrugated boxes, plastic bags, file envelopes, and other packaging materials (see Supplementary Table 9). These types are validated by our field investigation on package types and weights of over 8000 parcels in a number of distribution centers in various cities (see Supplementary Table 7).

We randomly gathered approximately 21,000 electronic waybills showing delivery origins and destinations, to determine the shares of various transportation modes. It is noted that we only collected transportation mode-related information via waybills provided by express delivery companies. We found that the transportation of parcels between distribution centers and regional centers (Phases II and IV) is mainly via road freight. The distance was determined using Baidu Map according to the actual road network. Distance between regional centers (Phase III) by air mode was estimated based on the routine airline distance of various airlines, and railway distance for Phase III was obtained from China Railway (see Supplementary Table 7).

GHG emissions factors corresponding to various transportation modes (truck, air and rail) and purchased electricity are from the Ecoinvent Life Cycle Inventory Database version 3.5, choosing China-specific data or data for comparable regions[36] (see Supplementary Table 10).

**Uncertainties.** Key parameters in this study include the shares of transportation modes for intercity deliveries, parcel weights, and shares of four categories of parcels. Normal distribution for parcel weights and uniform distribution for other parameters (e.g., fractions of transportation mode) were determined via statistical analysis. Data (parameters) from official statistical reports (e.g., annual delivery volume) and GHG emissions factors for transportation tools exported from the commercial database are set as accurate values (see Supplementary Table 11). Monte Carlo simulation with 10,000 samples was conducted to characterize uncertainties associated with these parameters. Sensitivities of the results to these parameters are also evaluated (see Supplementary Fig. 35).

Overall, both the total weight of scrap packaging materials and the transportation-related GHG emissions of express deliveries in 2018 follow normal distributions with a mean ($\mu$) of 8.8 MtCO$_2$e ($\sigma = 0.8$) (see Supplementary Fig. 36), and a mean ($\mu$) of 13.7 MtCO$_2$e ($\sigma = 2.2$) (see Supplementary Fig. 37), respectively. Normal distribution is also found for GHG emissions of intercity and intracity deliveries and of different parcel types (see Supplementary Fig. 38). The average deviation error for all the simulations was lower than 10% (9.1%), indicating that the model parameters were reasonable and controllable.

**Scenario analysis.** Based on the historical trend and future development of the express delivery sector, the logistic growth model is considered a reasonable approach to describe the growth of the volume of express deliveries, which was set as a constant parameter for conducting scenario analysis (see Supplementary Fig. 6)[22,37]. Meanwhile, five scenarios are developed to represent relevant strategies for GHG emission mitigation (Table 2). More justifications for scenario settings are shown in Supplementary Table 3.

## Data availability

The data that support the findings of this study are available in the Supplementary Information.

## Code availability

Please see supplementary materials for Monte Carlo simulation of GHG emission from the shipment of express delivery. Further code details can be obtained from the authors upon request.

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

## Acknowledgements

This work was financially supported by the Scientific Research Fund of Introduced High Talent of Shenzhen University (827-000044) and Natural Science Foundation of China (71991484). We are grateful to Prof. Qingyu Zhang, Prof. Jian Zuo, and Prof. Mingwei Hu for their valuable comments.

## Author contributions

H.D. and M.X. designed the research. P.K and G.S. conceived the paper, developed the model, and collected the data. P.K. ran the simulation and drew the figures. H.D., T.R.M., M.X., H.Z., and G.L. contributed to the conceptual model development. J.Z., Y.Z., H.W., J.R., and R.Z. enhanced scenario analysis discussions. All authors contributed to discussing the results and writing the paper.

## Competing interests

The authors declare no competing interests.
