## [Peer Review File · Nature Communications]

REVIEWER COMMENTS

Reviewer #1 (Remarks to the Author):

The idea is quite interesting and novel to explore the effects of packaging pollution of express delivery companies on environmental sustainability. There are sufficient references provided by the authors. Work appears convincing and the rationale is logical. I recommend the publication of your paper after following improvements:

1. Please add a brief conclusion of your research in a single paragraph, particularly highlighting your finding of the logistic growth model.
2. The environmental issues are strongly linked with the level of corporate social responsibility of the firms. You have given only a single reference for corporate social responsibility (Ref 22) in your paper. Please try to link your statements with the "employee level" factor of "corporate social responsibility" in your paper to strengthen the framework. What is the human factor of employees to save the environment from the damages being a micro-level CSR? How employees' satisfaction can foster it? For this purpose, I suggest you read and cite recent articles in your paper. For example, 2 useful articles for your paper are "Assessing human factor in the adoption of computer-based information systems as the internal corporate social responsibility (Farooq et al. 2019)" and "Unattended social wants and corporate social responsibility of leading firms: Relationship of intrinsic motivation of volunteering in proposed welfare programs and employee attributes (Hao et al., 2018)".
3. Please improve the English language of your paper. In the present form, it seems a little difficult to understand.

Overall, I am satisfied with your research and recommend its publication after minor revision.

Reviewer #2 (Remarks to the Author):

This study addresses an important issue which is appropriate for this journal. The research design is rigorous and the statistical analysis is appropriate and valid. The details provided for data collection and analysis help understand the research. The findings provide valuable insights to understanding the environmental impact (GHG emissions) of express deliveries and mitigation effects of some proposed strategies. Overall it is a well written paper. I have only a few minor concerns that require the authors to respond. The extremely high R2 makes me worrisome. We usually don't see such high R2. It is almost 100% correlation. Can you provide more explanation on this?

Some potential strategies on reducing GHG emissions of express deliveries were proposed by the authors. However, there lack sufficient justifications for these strategies. The justifications need to be provided.

The limitations and future studies should be included.

Responses to reviewers' comments

We greatly appreciate the opportunity to submit a revised manuscript along with our responses to two reviewers' comments. Many thanks to the two reviewers for providing the invaluable and constructive comments. The manuscript has been carefully revised, including fixing language and formatting errors. **We are providing both a clean and a track-change version of the revised manuscript.**

REVIEWER COMMENTS

Reviewer #1 (Remarks to the Author):

The idea is quite interesting and novel to explore the effects of packaging pollution of express delivery companies on environmental sustainability. There are sufficient references provided by the authors. Work appears convincing and the rationale is logical. I recommend the publication of your paper after following improvements.

Reply: We highly appreciate your positive comments. We have thoroughly updated the manuscript and shown more details of our study based on the comments of the editor and the two reviewers.

Please add a brief conclusion of your research in a single paragraph, particularly highlighting your finding of the logistic growth model.

Reply: We have taken the reviewer's advice. Specifically, we have added a conclusion section, including highlights of our findings from the logistic growth model.

Please see revised main text in lines 253-271 in pages 14-15.

The environmental issues are strongly linked with the level of corporate social responsibility of the firms. You have given only a single reference for corporate social responsibility (Ref 22) in your paper. Please try to link your statements with the "employee level" factor of "corporate social responsibility" in your paper to strengthen the framework. What is the human factor of employees to save the environment from the damages being a micro-level CSR? How employees'

satisfaction can foster it? For this purpose, I suggest you read and cite recent articles in your paper. For example, 2 useful articles for your paper are "Assessing human factor in the adoption of computer-based information systems as the internal corporate social responsibility (Farooq et al. 2019)" and "Unattended social wants and corporate social responsibility of leading firms: Relationship of intrinsic motivation of volunteering in proposed welfare programs and employee attributes (Hao et al., 2018)".

Reply: We agree that the environmental issues are strongly linked with the level of corporate social responsibility of the firms. The suggested references were carefully reviewed and indeed helped our analyses and have been cited in the revised paper. In the conclusion and limitation section, we explained that this concern is worth further study. "...the environmental impacts of an industry or a company probably strongly correlate with firm-level corporate social responsibility (CSR) efforts.^{2,728} An in-depth analysis of how various CSR initiatives or measures can affect the GHG emissions of the express delivery industry will be useful to guide companies to develop appropriate CSR strategies..."

Actually, we just completed a survey of consumers and other stakeholders related to the express delivery service to examine potential behavioral changes for GHG mitigation, which will be reported in a future publication.

Please see revised main text in lines 284-288 in pages 15-16.

Please improve the English language of your paper. In the present form, it seems a little difficult to understand.

Reply: The manuscript has been carefully revised, including fixing typographical and grammatical errors. A professional language editor from the U.S., Ms. Marian Rhys, has further revised the manuscript thoroughly.

Overall, I am satisfied with your research and recommend its publication after minor revision.

Reply: Thank you!

Reviewer #2 (Remarks to the Author):

This study addresses an important issue which is appropriate for this journal. The research design is rigorous and the statistical analysis is appropriate and valid. The details provided for data collection and analysis help understand the research. The findings provide valuable insights to understanding the environmental impact (GHG emissions) of express deliveries and mitigation effects of some proposed strategies. Overall, it is a well written paper. I have only a few minor concerns that require the authors to respond.

Reply: Thank you!

The extremely high R^2 makes me worrisome. We usually don't see such high R^2 . It is almost 100% correlation. Can you provide more explanation on this?

Reply: Thank you for the careful observation. We double-checked our data and analyses, and confirm the high R^2 for the stepwise regression of GHG emissions from express deliveries with various factors, including the road transportation distance per piece (x_1) and the urbanization rate (x_2). We provide the explanation as follows.

First, we used the European Norm EN 16258 methodology to estimate energy consumption and GHG emissions for transportation. European Norm EN 16258 is a standardized methodology for the calculation and declaration of energy consumption and GHG emissions related to any transportation operation. **The key parameter for this method is the transportation distance.**

Second, in our stepwise regression, we indeed found a high correlation between the transportation distance and the GHG emissions per piece delivered for all three transportation modes. The correlation coefficient is more than 93% with high statistical significance, as shown in Table R1 in this document. This is the main reason for the high correlation in the results.

In order to reduce the collinearity among the distance of the three transportation modes, the value with the highest correlation (road transportation distance per piece) is selected as the explanatory variable in the stepwise regression model.

We have added these discussions in the revised manuscript.

The main reason was the key parameter for this method is the transportation distance, especially the road transportation distance per piece.

Table R1 Correlation between the distance of the three transportation modes and the GHG emissions per piece

Index	Correlation coefficient value	t value	Significance level P
Road transportation distance per piece	0.959	21.496	<2.2E-16
Air transportation distance per piece	0.932	16.346	<2.2E-16
Railway transportation distance per piece	0.949	19.065	<2.2E-16

Some potential strategies on reducing GHG emissions of express deliveries were proposed by the authors. However, there lack sufficient justifications for these strategies. The justifications need to be provided.

Reply: We agree with this suggestion. We have provided more concrete justifications for these strategies in the revised manuscript (also in Supplementary Table S3, or Table R2 here).

Table R2 Justifications for scenario analysis

Scenario	Background and significance	Relevant measures
S1 (Faster delivery)	S1 and S2 are justified by the Chinese government's plan for the express delivery industry ¹² in which diversified modes of express delivery services are	Increasing the proportion of aircraft transportation could increase delivery speed for S1. Specifically, the number of dedicated cargo planes increased from 57 in 2013 to

Scenario	Background and significance	Relevant measures
	encouraged including both faster delivery speed and slower one to serve different consumer demands.	116 in 2018 in China. ^{13,14} So far, three express delivery companies have their own airline fleets. ¹² It is true that delivery time is becoming shorter and shorter; this is a key indicator to evaluate the quality and competitiveness of an e-commerce company. For example, the average delivery time by ‘Tmall’ (one of the top e-commerce companies in China) has fallen from 9 days in 2013 to 2.8 days in 2018. ¹⁵
S2 (Slower delivery)		Many deliveries are not urgent, thus do not necessarily need fast delivery. Slower, cheaper, and also less carbon-intensive delivery services are preferred for such demand, such as delivery by railways. ¹⁶ Express delivery companies are indeed developing railway-based express delivery services. ¹⁷
S3 (Fuel standard upgrade)	Fuel standard upgrade is a nation-wide policy that affects all transportation-related sectors including the express delivery sector (National Development and Reform Commission (NDRC) Announcement on Oil Products Upgrading-Announcement No.16. ¹⁸)	The National Development and Reform Commission (NDRC) in China has issued more stringent standards leading to the implementation of China VI standards (Euro 4) in 2020.
S4 (Less packaging material)	China’s State Council issued the “Express Delivery Interim Regulations” in 2019 including a mandate to reduce packaging materials for express deliveries by 15% through means such as improved source control and using environmentally friendly	Along with the release of the revised version of the "Packing Standard for Express Service" (GB/T 16606-2018) in 2018 ¹⁸ which sets higher environmental standards for packaging products, the State Post Bureau has begun to vigorously promote the

Scenario	Background and significance	Relevant measures
	packaging products.	implementation of the “9571 program**” which targets at reducing the amount of packaging materials. ¹⁷ Express delivery companies have invested significantly in environmentally friendly packaging, such as the Alibaba Green Logistics 2020 plan, the Jingdong (JD)-Qingliu plan, the Suning-Qingcheng plan, and the SF express-Fengjing plan. ¹⁹
S5 (Logistics optimization)	The transportation and logistics network is one of the most important asset of express delivery companies. Constantly optimizing the network by increasing delivery volume and speed is a major undertaking for the companies to reduce cost and improve custom satisfaction. As a result, it also reduce GHG emissions and other environmental imapcts. ²⁰	Express delivery companies are constantly optimizing their transportation and logistics networks, by introducing innovative technologies (such as cloud computing, big data, artificial intelligence, and blockchain). ²¹

Note: References can be found in Table S11 in Supporting Information file.

The limitations and future studies should be included.

Reply: We summarized the limitations of the research from the aspects of the ‘big data’ features, system boundary, and the major trends of the express delivery industry. We also put forward suggestions for future study.

Please see revised main text in lines 272-288 in pages 15-16.

** The “9571 program” means: the proportion of electronic waybill will account for 95% of the total, over 50% of the parcels will not use external packaging materials for delivering purposes, the reusable transshipment packaging bags will reach 70% of the total, and more than 100,000 express delivery service centers will recycle packaging materials.*